# Optimization of the Black Garlic Processing Method and Development of Black Garlic Jam Using High-Pressure Processing

**DOI:** 10.3390/foods12081584

**Published:** 2023-04-08

**Authors:** Wen-Chang Chang, Wen-Chun Lin, She-Ching Wu

**Affiliations:** Department of Food Sciences, National Chiayi University, Chiayi 600355, Taiwan; wcchang@mail.ncyu.edu.tw (W.-C.C.);

**Keywords:** black garlic, high-pressure processing, antioxidant capacity, Maillard reaction, jam, texture analysis

## Abstract

Black garlic has many beneficial effects, and it has a less spicy flavor. However, its aging conditions and related products still need to be further investigated. The present study aims to analyze the beneficial effects under different processing conditions and utilize high-pressure processing (HPP) in the production of black garlic jam. The highest antioxidant activities, including the DPPH scavenging, total antioxidant capacity, and reducing power (86.23%, 88.44%, and A_700_ = 2.48, respectively), were observed in black garlic that had been aged for 30 days. Similarly, the highest total phenols and flavonoids were observed in black garlic that had been aged for 30 days (76.86 GAE/g dw and 13.28 mg RE/g dw, respectively). The reducing sugar in black garlic was significantly increased to about 380 (mg GE/g dw) after 20 days of aging. The free amino acids in black garlic were decreased time-dependently to about 0.2 mg leucine/g dw after 30 days of aging. For the browning indexes of black garlic, the uncolored intermediate and browning products were increased in a time-dependent manner and reached a plateau at day 30. Another intermediate product in the Maillard reaction, 5-hydroxymethylfurfural (5-HMF), was observed in concentrations that increased to 1.81 and 3.04 (mg/g dw) at day 30 and 40, respectively. Furthermore, the black garlic jam made by HPP was analyzed for its texture and sensory acceptance, showing that a 1:1.5:2 ratio of black garlic/water/sugar was the most preferred and was classified as “still acceptable”. Our study suggests suitable processing conditions for black garlic and outlines the prominent beneficial effects after 30 days of aging. These results could be further applied in HPP jam production and increase the diversity of black garlic products.

## 1. Introduction

Garlic is used as a food ingredient, spice, and herbal medicine because of its bioactive compounds and beneficial effects [1,2]. Several biological functions of garlic, such as its antioxidant activities, cardiovascular protection, anti-obesity effect, and neuroprotection, have been documented in a literature review and suggested to be related to bioactive compounds, such as organosulfur compounds and polyphenols [2]. In addition, the potential amelioration of garlic on neurodegeneration disease models such as Alzheimer’s disease and Parkinson’s disease has also been reported [3,4]. The spicy and intensive odor of freshly crushed garlic is due to allicin, one of the organosulfur compounds catalyzed by alliinase [5]. Allicin can modulate the redox balance in cells and cellular signaling, which might be associated with the antioxidant activity of garlic [6]. However, the pungent flavor which originates from allicin might limit the consumption of fresh garlic [7,8], highlighting the need for alternative garlic products without an intensive odor.

Black garlic is an aged garlic product which is aged under a regular temperature (60–90 °C) and humidity (70–90%) and possesses a unique, sweet, and less-spicy taste and a less intense odor compared with fresh garlic [9]. Despite the reduced allicin content, black garlic has higher antioxidant activities compared with fresh garlic, which might be due to the higher *S*-allyl cysteine (SAC) that transforms from allicin during aging [8,10]. In addition, higher contents of polyphenols and flavonoids were observed in black garlic compared with fresh garlic [11]. The color and odor of black garlic are attributed to the Maillard reaction during aging [12], and Maillard products, such as 5-hydroxymethylfurfural (5-HMF), are suggested to be anti-inflammatory and vascular protective, and also to possess other health effects [8,13]. The health effects of black garlic, such as its anti-inflammatory, anti-diabetes, and anti-cancer properties, neuron and cardiovascular protection, and immunoregulation, have also been reported [8,10]. Moreover, altered aging conditions, including temperature and humidity, can cause different antioxidant activities in black garlic [14,15], suggesting the importance of controlling the aging conditions.

For better food preservation without additives or thermal processing that might influence the flavor, high-pressure processing (HPP) has been applied in the production of fermented foods and meat products [16,17]. In addition, research using HPP in garlic or aged garlic products (e.g., Laba garlic) suggests that the beneficial effects are retained, and that the products have sensory acceptability [18,19]. An accelerated Maillard reaction in black garlic aging under HPP was also reported and showed commercial potential [20]. However, the influence of the aging conditions of black garlic on its beneficial effects and the feasibility of producing black garlic jam using HPP are not yet fully understood. Herein, our study aims to investigate the antioxidant activity, browning index, and bioactive compounds to determine the better aging conditions for black garlic. We have further utilized the HPP technique to produce black garlic jam and conduct a texture analysis and sensory evaluation for its commercial application. Our results could help in the production of black garlic with more beneficial effects, and our results suggest the commercial feasibility of producing black garlic jam.

## 2. Materials and Methods

### 2.1. Chemicals

Adenosine and cordycepin were purchased from Sigma Chemical Co. (St. Louis, MO, USA). The hematoxylin and eosin (HE) stain kit (CATA: ab245880) and periodic acid Schiff (PAS) stain kit (CATA: ab150680) were purchased from Abcam (Cambridge, MA, USA).

### 2.2. Black Garlic Sample Preparation

Fresh garlic was purchased from a wholesale market in Yulin County, Taiwan. The aging of the black garlic was conducted in a chamber under a constant temperature (70 °C) and humidity (80%). The black garlic was sampled at day 5, 10, 20, 30, 40, and 45 and it was stored at 4 °C for the following experiments. Subsequently, the black garlic was treated under 10-fold volume deionized water in a 100 °C water bath for 60 min for the preparation of hot-water extracts. The extracts were centrifuged (4000 rpm) at 25 °C for 10 min and filtered using Whatman#1 filter paper and a suction filter. The residue was then processed again using the same extraction method and the filtrate was mixed together for the preparation of the final black garlic hot-water extracts. The fresh garlic was extracted in the same way to be compared with black garlic. The physicochemical properties, including the color, pH, moisture, and water activity, were analyzed using a colorimeter (Solorimeter ZE-2000, Nippon Denshoku Industries Co., Ltd., Tokyo, Japan), pH meter, and moisture analyzer (HB43 Halogen moisture analyzer, Mettler Toledo Inc., Columbus, OH, USA), respectively.

### 2.3. Antioxidant Activities and Bioactive Compounds Analyses

The antioxidant activities and bioactive compounds, including the DPPH scavenging ability and Trolox equivalent antioxidant capacity (TEAC), reducing power, total phenols, and flavonoids, were analyzed in black garlic samples aged for different periods. A detailed description of the methods which were used is as follows:

The DPPH assay was conducted according to a previous study [21]. Briefly, 1 mL of black garlic extracts were mixed with 5 mL of DPPH solution (0.1 mM in methanol). After a 50 min reaction in the dark, the optical absorbance at 517 nm (A_517_) of the mixture was measured. The DPPH scavenging effects were calculated as:(1)$$\mathrm{DPPH}\;\mathrm{scavenging}\;\mathrm{effect}\;\left(\%\right)=\frac{A_{517,\;\mathrm{blank}}-A_{517,\;\mathrm{sample}}}{A_{517,\;\mathrm{blank}}}\times 100\%$$

The TEAC was measured by ABTS^+^ scavenging adapted from a previous study [22]. The ABTS^+^ solution was prepared by mixing 1.5 mL of deionized water, 0.25 mL of peroxidase (44 units/mL), 0.25 mL of H_2_O_2_ (500 μM), and 0.25 mL of ABTS (1000 μM) and incubating the solution in the dark for 1 h. The black garlic extracts (0.25 mL) were subsequently added to the ABTS^+^ solution, and the optical absorbance was measured at 734 nm after 1 min. The TEAC was calculated as:(2)$$\mathrm{TEAC}\;\left(\%\right)=\left(1-\frac{A_{734,\;\mathrm{sample}}}{A_{734,\;\mathrm{blank}}}\right)\times 100\%$$

The reducing power was analyzed by mixing 1 mL of fresh garlic or black garlic extracts, 0.8 mL of phosphate buffer (pH 6.6, 0.2 M), and 1 mL of potassium ferricyanide (1%) in a 50 °C water bath for 20 min. After it had been cooled by ice water, to the mixture we added 1 mL of 10 % trichloroacetic acid, and the mixture was then centrifuged (3000 rpm) for 10 min. Then, 1 mL of supernatant was mixed with 1 mL of deionized water and 0.1 mL of ferric chloride (0.1%) for 10 min. The optical absorbance at 700 nm was measured, and this represents the reducing power [23].

The measurement of the total phenols was conducted based on previous studies [24,25]. Briefly, 0.1 mL of fresh garlic or black garlic extracts were mixed with 1 mL of Na_2_CO_3_ (2%) for 4 min, and then we added 0.5 mL of Folin–Ciocalteu’s reagent (50%) in the absence of light for 1 h. The optical absorbance at 750 nm of the mixture was measured and calibrated using various concentrations of gallic acid.

The flavonoids were analyzed according to Kubola and Siriamornpun [26]. The fresh garlic or black garlic extracts (0.5 mL) were mixed with 2.25 mL of deionized water and 0.15 mL of NaNO_2_ (5%). After reacting for 6 min, 0.3 mL of AlCl_3_·6 H_2_ O (10%) was mixed in and left to stand for 5 min in the dark. Then, 1 mL of NaOH (1 M) was added to the mixture and we measured the optical absorbance at 510 nm using various concentrations of rutin for the calibration.

### 2.4. Browning Reaction Analyses

The reducing sugar in fresh garlic and black garlic was measured using a dinitrosalicylic acid (DNS) reagent [27]. Briefly, 1 mL of garlic extract was mixed with 1 mL of DNS reagent and was incubated at 85 °C for 10 min. Subsequently, the mixture was cooled immediately, and we then added 10 mL of deionized water. The optical absorbance at 540 nm was then measured and calibrated with 0–1 mg/mL of glucose.

The free amino acids were measured using 2,4,6-trinitrobenzenesulfonic acid (TNBS) [28]. The fresh garlic and black garlic powder was dissolved in 1% SDS (sodium dodecyl sulfate) solution. Briefly, 40 μL of the sample and leucine standard solution, 320 μL of phosphate buffer (2M, pH 8), and 320 μL TNBS (0.1%) were mixed together and incubated in a 50 °C water bath for 60 min in the absence of light. After the sample had cooled at room temperature, 640 μL of HCl (0.1N) was added along with a 200 μL aliquot to a 96-well plate for the measurement of the optical absorbance at 340 nm. The concentrations of free amino acids were calculated based on the standard curve of leucine.

The products of the Maillard reaction were analyzed according to a previous study with a slight modification [29,30]. The fresh garlic and black garlic extracts were adjusted to 1 mg/mL using deionized water, and 200 μL aliquots were added to a 96-well plate to determine the optical absorbance at 294 nm and 420 nm.

The 5-HMF in the fresh garlic and black garlic extracts was determined using HPLC (L-7200, Hitachi Ltd., Tokyo, Japan), and this was compared with 5-HMF standards [31]. The analytical conditions were as follows. Column: Mightysil RP-C18 (250 mm × 4.6 mm × 5 μm); velocity: 1 mL/min; detector: Hitachi L-7455 diode array detector; wavelength: 280, 320, 370, and 520 (nm); inlet: 20 μL; mobile phase: A solution (2 acetic acid) and B solution (0.5% acetic acid-acetonitrile, 1:1 *v/v*); and gradient: 0 min (A:B = 90:10), 20 min (A:B = 90:10).

### 2.5. Preparation of Black Garlic Jam, Texture Analysis, and Sensory Evaluation

Various ratios of black garlic, water, and sugar (1:1.5:2, 1:2:1, 1:1:2, 1:1:1, 1:2:1.5, and 1:1.5:2) (*w/w/w*) were used to produce black garlic jam. After mixing the black garlic, water, and sugar, the pH was adjusted to pH 3.4 using food-grade citric acid, and 1.5% high-methoxyl pectin, which had been extracted from citrus, was added. The products were treated under high pressure (300 MPa) for 15 min to obtain black garlic jam.

The texture profile analysis was conducted using a back extrusion set to compress the samples twice. The compressed samples were added to a container with a 5 cm diameter and they were pressed (1 mm/s, 40%, trigger force = 5 g) to analyze the hardness, fracturability, chewiness, adhesiveness, cohesiveness, springiness, resilience, and gumminess.

The sensory evaluation was conducted according to Fiorentini et al. [32]. Several non-trained evaluators (age: 18–50, *n* > 30) scored the black garlic jam samples for their color, aroma, sweetness, sourness, bitterness, texture, smoothness, and overall preference using a 9-point scale. On the scale, 9 points represented the highest preference and acceptance, whereas 1 point represented the lowest, and 5 points was the midpoint.

### 2.6. Statistical Analysis

The results were presented as mean ± standard deviation (SD). SPSS (Statistical Product and Service Solutions 22.0) was used for the data analysis, and a one-way ANOVA was performed for comparisons between the groups. Duncan’s post hoc multiple range method was used to compare significant differences, and *p* < 0.05 was regarded as statistically significant.

## 3. Results and Discussion

### 3.1. Appearance and Physicochemical Properties of Aged Black Garlic for Different Aging Times

Figure 1A shows the appearance of fresh garlic and black garlic at an aging duration which is better for its preservation. The pH of the black garlic extracts decreased with the aging duration from pH 5.72 to 3.74 (Figure 1B). Heating treatment can increase the acidity of garlic, as the formation of carboxylic acid leads to an increase in acidity and a decrease in pH. It has also been reported that in the process of making black garlic, organic acids are released by breaking down the cells, and organic acids are also generated during the heating process, leading to a decrease in the pH value of black garlic [33]. Figure 1C and Table 1 show that the longer aging durations decreased the moisture content and the water activity of the black garlic samples. The yield of extract from black garlic increased from 63.8 to 68.9% with an increase in aging duration from 5 to 20 days, but a decrease in the yield was also observed. For comparison, the yield of extract from fresh garlic was 33.7% (Table 1). During the maturation process of black garlic, the Maillard reaction occurs between the carbonyl group of reducing sugars and the amino groups of free amino acids, peptides, and proteins, leading to complex chemical reactions and the formation of black melanoidins [33]. The color analysis shows the decreased brightness (L value) and colors (red and yellow; a and b values) (Table 1).
Extract yield (%) = dry weight of different aging day extract/dry weight of sample × 100%ΔE = [(L − L_0_)^2^ + (a − a_0_)^2^ + (b − b_0_)^2^]^1/2^(3)

### 3.2. Antioxidant Activity of Fresh Garlic and Black Garlic Samples

The DPPH scavenging ability, TEAC, and reducing power of fresh garlic and black garlic hot-water extracts were analyzed to determine the antioxidant activity of fresh garlic and black garlic. Figure 2A illustrates that the DPPH scavenging effect (5 mg/mL) of fresh garlic was 11.7% only. For aged black garlic, this effect increased from 40.4 to 86.2% with the increase in aging duration from 5 to 30 days, but with further aging, it slightly decreases. A previous study reported that under 70 °C and 90% humidity, the DPPH scavenging effect of fresh garlic was 4.65%, and this increased to 37.32%, 74.48%, and 63.09% after 7, 21, and 35 days of aging [11], which is similar to our results (Figure 2A). The ethyl acetate extracts of black garlic were shown to have >70% DPPH scavenging effects in 100 μg/mL, and various compounds were characterized, such as 5-HMF, that could potentially decrease oxidative stress in HepG2 cell assays [34], indicating the antioxidant role of Maillard products.

The TEAC of the fresh garlic and black garlic samples was analyzed using ABTS^+^ assays. The results show that the TEAC (5 mg/mL) of fresh garlic was 25.3%. The TEAC of aged black garlic increased from 33.2 to 88.4% with the increase in aging duration from 5 to 30 days and then decreased with further aging (Figure 2B). In addition, the highest TEAC and DPPH scavenging effect were both observed after 30 days of aging. There was a significant difference in TEAC and DPPH scavenging between aged black garlic and fresh garlic (*p* < 0.05). Similarly, the ABTS^+^ scavenging ability of black garlic increased as the aging time increased, and it showed the highest effects at day 21 and 28 under 70 °C and 90% humidity [11]. The reducing power analyzed by the ferricyanide reduction revealed a time-dependent increase in black garlic samples, whose optical absorbance at 700 nm was 0.76–2.48 (Figure 2C). The results show that the highest reducing power was observed in black garlic aged for 20 and 30 days (Figure 2C), which is similar to the DPPH and TEAC results. Similar results were reported in another study which found that the aging of black garlic for 21 and 28 days at 70 °C had the highest reducing power [11]. Previous research has suggested that 85 °C and 45 days of aging resulted in the highest reducing power of black garlic (OD700 = 3.13) [35], suggesting the effects of temperature and duration on the antioxidant activities of black garlic. In summary, our results show the significant increase in the antioxidant activities of black garlic compared with fresh garlic, and the highest increases were mostly in the 30-day aging group.

### 3.3. Bioactive Compounds in Fresh Garlic and Black Garlic Extracts

Table 2 shows that the highest total phenols were found in black garlic compared with fresh garlic and that there was a time-dependent increase in the first 30 days. Total phenols (at mg GAE/g dw) in the extract of black aged garlic increased from 10.9 to 74.9% with the increase in aging duration from 5 to 30 days but decreased with further aging. In addition, flavonoids are associated with total phenols and show similar trends, with 0.57 in fresh garlic and 1.34–13.28 in black garlic (mg RE/g dw) and the highest levels at day 30. The content of flavonoids in the hot water extract of black garlic made after 20 days of maturation showed no significant change. The contents of total phenols and flavonoids in aged black garlic were significantly higher than in fresh garlic (*p* < 0.05). Despite the fact that the phenolic compound contents might be affected by species, climates, and seasons [36], our results suggest the significant increase in the total phenols resulted from the aging of black garlic. In a previous study, the black garlic aged under 70 °C and 90% humidity possessed the highest total phenols after 21 days (58.33 mg GAE/g dw) [11], which is in agreement with our results (Table 2). In addition, previous research has outlined the effects of the processing conditions on black garlic and the higher flavonoids in black garlic compared with fresh garlic [11,14].

### 3.4. Browning Reaction Analyses for Black Garlic

The browning reaction analyses, including reducing sugar, free amino acids, browning indexes, and 5-HMF determination, were conducted on black garlic which had been aged for different durations. The results show that the reducing sugar was 3.24 (mg GE/g dw) in fresh garlic, while the reducing sugar in the extract of black aged garlic increased from 10 to 39 as the aging duration increased from 5 to 30 days, but decreased with further aging (Figure 3A). Our results suggest that there was an increase in the reducing sugar in black garlic throughout the aging duration up until day 40, and there was no significant increase after day 20. Similarly, Choi et al. (2014) demonstrated in their study that the reducing sugar in black garlic increased from 1.2 to 16.07 (g/kg), and no significant difference was observed after 21 days [11]. In addition, a lower content of reducing sugar was found in black garlic under 90 °C aging compared with <70 °C, but this was accompanied by a poor quality due to intensive burning, resulting in a more intense odor and a tough texture [33].

The free amino acids in fresh garlic and black garlic samples were in the range 0.18–0.58 (mg leucine/g dw), and the contents decreased throughout the aging duration (Figure 3B). Leucine is the precursor of organosulfur compounds which can be transformed into other volatile compounds [7,37]. In addition, it has been suggested that the decreased amino acids were related to the changed odor and antioxidant activities of black garlic [11,30].

The browning reaction analyses were measured by the brown index, as shown in Figure 4A; 5-HMF is the crucial intermediate and an indicator of the Maillard reaction. Figure 4B depicts the HPLC chromatogram of fresh garlic and black garlic extracts. Table 3 shows that 5-HMF was only detected in black garlic aged for 20, 30, 40, and 45 days, and the contents were 0.12, 1.81, 3.04, and 3.43, respectively (mg 5-HMF/g dw). Increases in 5-HMF concentration with aging temperature and duration were also reported in another study, and 70 °C was found to be suitable for increasing the rate of 5-HMF in black garlic production [33]. In addition, under 70 °C, the 5-HMF in black garlic was barely detected before day 12 [33], which is consistent with our results (Table 3). In summary, the results from the browning reaction analysis suggest that suitable levels of reducing sugar and the browning reaction were achieved at day 30. Moreover, the antioxidant activities were the highest in black garlic after 30 days of aging. Therefore, black garlic which had been aged for 30 days was further used to produce black garlic HPP jam using different ratios of black garlic, water, and sugar.

### 3.5. Sensory Evaluation of Black Garlic Jam by HPP Treatment

We used black garlic which had aged for 30 days under conditions of 70 °C and 80% humidity to produce HPP jam using 300 MPa high-pressure processing for 15 min. Figure 5A presents the appearance of the HPP jam produced using different ratios of black garlic, water, and sugar. A previous study has shown that the consumer acceptance values for jam texture are as follows: hardness of 58.67–124.97 g, elasticity of 0.92–0.94, adhesiveness of 66.65–557.48 g·, cohesion of 0.79–0.89, gumminess of 46.78–98.55 g, and chewiness of 46.61–90.68. These values indicate that the texture of the jam is slightly soft and sticky [38]. The results of the texture analysis show the hardness, fracturability, adhesiveness, springiness, chewiness, gumminess, cohesiveness, and resilience (Table 4). The acceptable hardness level and better fracturability for the 1:1.5:2 ratio (black garlic/water/sugar) can be observed in Table 4. The 1:1.5:2 and 1:1:2 ratios show the relatively low but acceptable values of springiness due to the higher sugar content (Table 4). The chewiness and gumminess also indicate the commercial acceptability of the 1:1.5:2 samples (Table 4). In addition, the 1:1.5:2, 1:1:1, and 1:1.5:1 ratios show acceptable cohesiveness, and the 1:2:1.5 sample displays the best resilience (Table 4). We then conducted further sensory evaluations of black garlic jam. The sweet, smooth, and texture scores for the 1:1.5:2 sample were obviously higher than for the other samples (Figure 5B). Lower bitter and scour scores were also observed in for samples with the 1:1.5:2 ratio. Overall, the acceptance was mostly deemed “still acceptable”. In summary, black garlic HPP jam with the ratio of 1:1.5:2 (black garlic/water/sugar, *w/w/w*) received the highest preference.

HPP has been used to kill harmful pathogens and prevent food spoilage without resulting in heat-caused transformation of chemicals [39]. In addition, HPP results in different sensory properties, such as flavor and texture, but the effects of high pressure are still unclear [40]. Several previous studies used HPP to produce fresh garlic paste and black garlic, which showed a different texture and sensory properties [19,20]. Our results further investigated black garlic HPP jam associated with water and sugar under different ratios, revealing the effects of HPP on the texture of black garlic jam. In addition, HPP might improve the sensory attributes of black garlic compared with heat treatment [19]. Our results suggest different preferences for black garlic HPP jam with different ratios of black garlic, water, and, sugar.

## 4. Conclusions

Our results show that black garlic hot-water extracts have better antioxidant activities compared with fresh garlic (5 mg/mL), especially after 30 days of aging, in terms of the DPPH scavenging effect, TEAC, reducing power, and bioactive compounds, such as total phenols and flavonoids. The moisture and water activity of black garlic aged after 30 days was relatively low, which improves its preservation. In addition, the reducing sugar, browning index, and 5-HMF results suggest a better browning reactive level in black garlic that has been aged for 30 days. Furthermore, better texture and improved sensory properties of black garlic HPP jam were observed at a ratio of 1:1.5:2 (black garlic/water/sugar, *w*/*w*/*w*). Taken together, our results suggest that optimal physicochemical and antioxidant properties of black garlic can be achieved under the processing conditions of 70 °C, 80% humidity, and 30 days of aging. The present study also provides information on the preferences for black garlic HPP jam, which is helpful for its further commercial applications.

## Figures and Tables

**Figure 1 foods-12-01584-f001:**
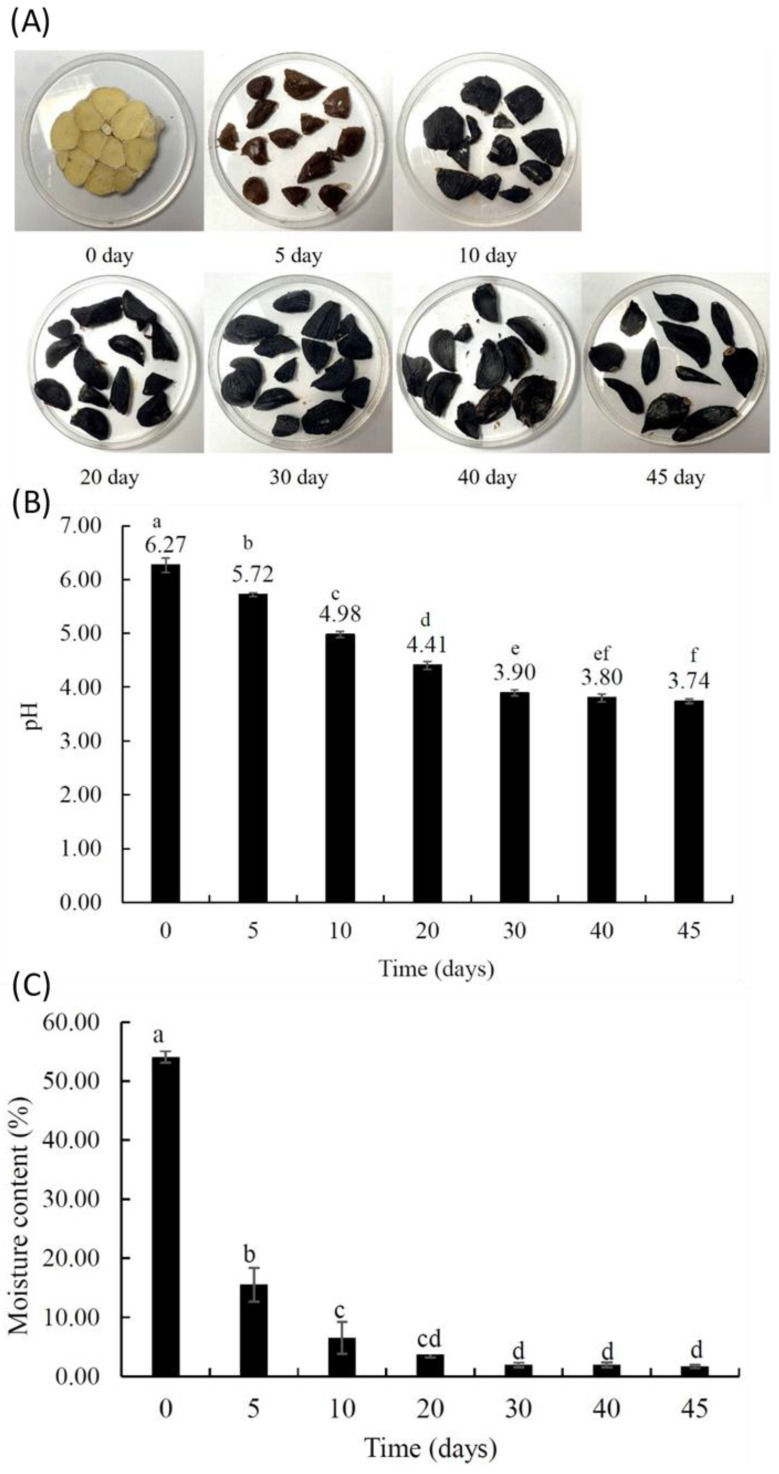
(**A**) The appearance of fresh garlic and black garlic obtained from different aging durations. (**B**) pH value of hot-water extract broth of fresh garlic and black garlic obtained from different aging durations. (**C**) Moisture content of fresh garlic and black garlic obtained from different aging durations. Each value is expressed as mean ± S.D. (*n* = 3). Different superscript letters indicate a significant difference (*p* < 0.05).

**Figure 2 foods-12-01584-f002:**
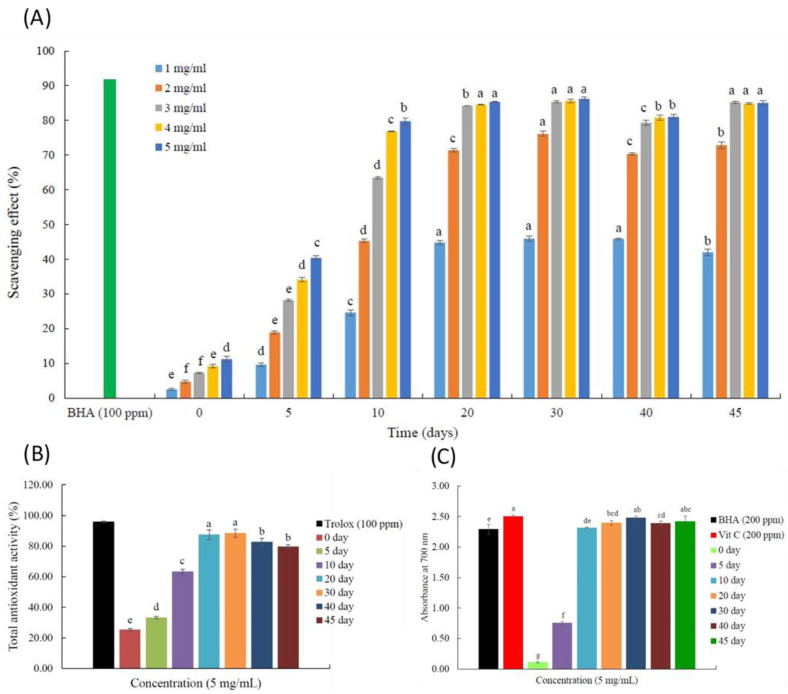
(**A**) DPPH radical scavenging effect, (**B**) total antioxidant activity, and (**C**) reducing power of hot-water extracts obtained from fresh garlic and black garlic with different aging durations. (**B**) Values are shown as the mean ± S.D. (*n* = 3). Different superscript letters indicate a significant difference (*p* < 0.05).

**Figure 3 foods-12-01584-f003:**
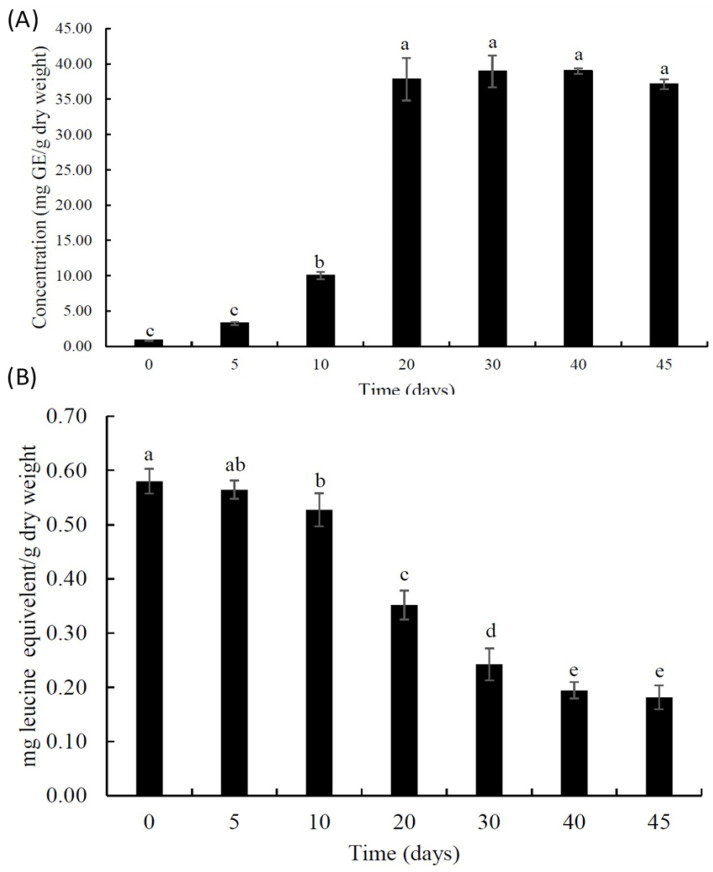
(**A**) Reducing sugar and (**B**) free amino acid contents of hot-water extract obtained from fresh garlic and black garlic with different aging durations. Values are shown as the mean ± S.D. (*n* = 3). Different superscript letters indicate a significant difference (*p* < 0.05). GE: glucose equivalent.

**Figure 4 foods-12-01584-f004:**
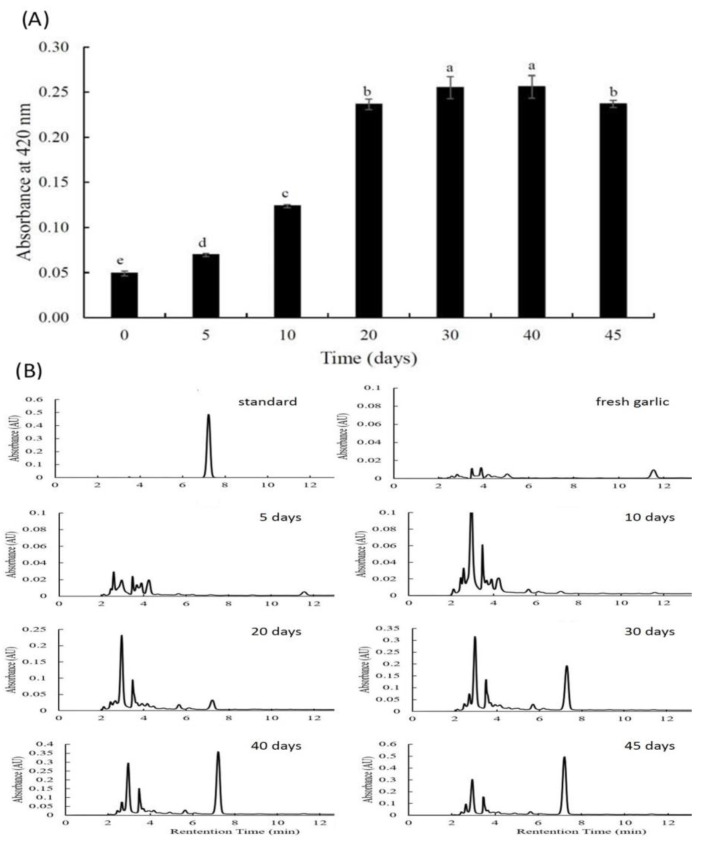
(**A**) The brown index and (**B**) HPLC chromatogram of hot-water extracts obtained from fresh garlic and black garlic for 5-HMF for different aging durations. Values are shown as the mean ± S.D. (*n* = 3). Different superscript letters indicate a significant difference (*p* < 0.05).

**Figure 5 foods-12-01584-f005:**
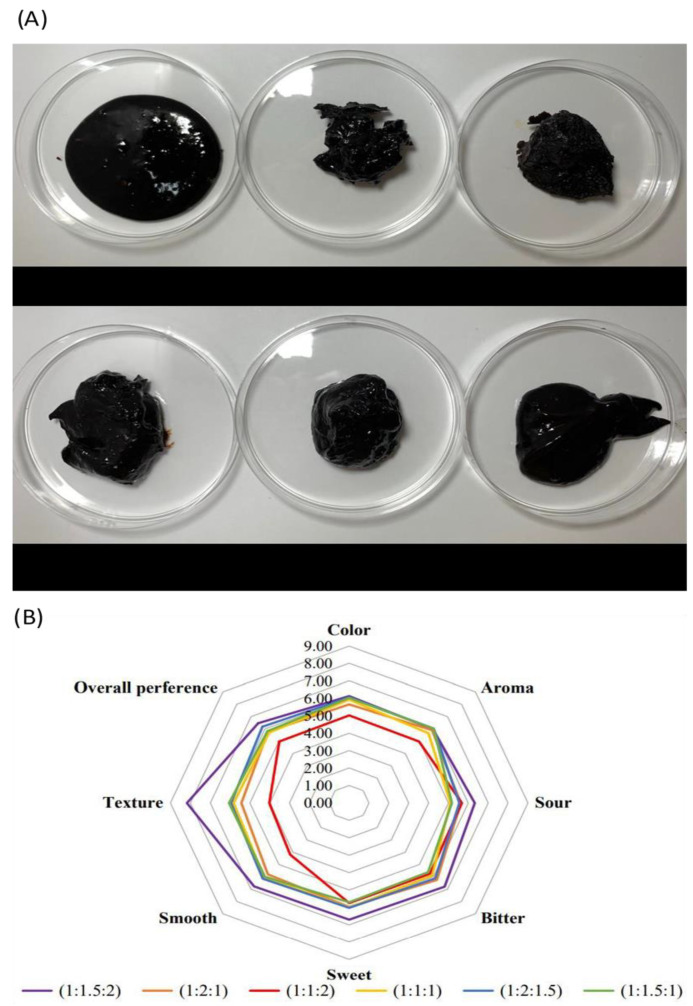
(**A**) The appearance of black garlic jam with different ratios produced by HPP treatment. (**B**) The sensory evaluation of black garlic jam with different ratios by HPP treatment. Six different ratios of garlic/water/sugar (*w/w*) were tested: black garlic/water/sugar (*w/w*) = 1:1.5:2; black garlic/water/sugar (*w/w*) = 1: 2:1; black garlic/water/sugar (*w/w*) = 1: 1:2; black garlic/water/sugar (*w/w*) = 1: 1:1; black garlic/water/sugar (*w/w*) = 1: 2:1.5; and black garlic/water/sugar (*w/w*) = 1:1.5:1.

**Table 1 foods-12-01584-t001:** Extract yield, water activity, and color of fresh garlic (on day 0) and black garlic obtained from different aging days.

Time (Days)	Extract Yield (%)	Water Activity (A_w_)	Hunter Color Values			
			L	a	b	ΔE
0	33.69 ± 1.47 ^c^	0.97 ± 0.003 ^a^	59.98 ± 0.80 ^a^	−1.35 ± 0.47 ^c^	17.57 ± 1.06 ^a^	-
5	63.85 ± 2.87 ^ab^	0.88 ± 0.011 ^b^	18.34 ± 2.40 ^b^	6.86 ± 0.53 ^a^	8.25 ± 1.34 ^b^	43.45
10	67.36 ± 2.49 ^a^	0.75 ± 0.012 ^c^	16.14 ± 1.62 ^c^	1.37 ± 0.65 ^b^	1.76 ± 1.27 ^c^	46.68
20	68.93 ± 2.67 ^a^	0.69 ± 0.007 ^d^	15.92 ± 1.00 ^c^	−0.29 ± 0.96 ^c^	0.40 ± 0.53 ^c^	47.3
30	58.43 ± 3.41 ^b^	0.70 ± 0.012 ^d^	12.76 ± 1.08 ^d^	0.96 ± 0.53 ^b^	1.49 ± 0.77 ^c^	49.93
40	52.71 ± 2.09 ^c^	0.60 ± 0.010 ^e^	13.59 ± 1.35 ^d^	0.54 ± 0.85 ^b^	0.36 ± 0.63 ^c^	49.52
45	49.23 ± 3.15 ^c^	0.58 ± 0.006 ^e^	13.63 ± 1.21 ^d^	−0.25 ± 0.33 ^bc^	0.34 ± 0.33 ^c^	49.7

Values are shown as the mean ± S.D. (*n* = 3). Different superscript letters indicate a significant difference (*p* < 0.05).

**Table 2 foods-12-01584-t002:** Total phenolics and total flavonoids contents of hot-water extracts obtained from fresh garlic and black garlic with different aging durations.

Time (Days)	Contents	
	Total Phenolics	Total Flavonoids
	(mg GAE/g Dry Weight)	(mg RE/g Dry Weight)
0	6.73 ± 0.67 ^g^	0.57 ± 0.3 ^c^
5	10.87 ± 0.57 ^f^	1.34 ± 0.24 ^c^
10	28.35 ± 0.31 ^e^	4.79 ± 0.95 ^b^
20	58.24 ± 0.89 ^b^	12.28 ± 0.78 ^a^
30	74.86 ± 1.13 ^a^	13.28 ± 0.51 ^a^
40	50.75 ± 0.41 ^c^	12.85 ± 0.66 ^a^
45	46.79 ± 0.42 ^d^	12.91 ± 0.31 ^a^

Values are shown as the mean ± S.D. (*n* = 3). Different superscript letters indicate a significant difference (*p* < 0.05). GAE: gallic acid equivalent. RE: rutin equivalent.

**Table 3 foods-12-01584-t003:** The 5-HMF content of hot-water extracts obtained from fresh garlic and black garlic with different aging durations.

Time (Days)	5-HMF
	(mg/g Dry Weight)
0	N.D.
5	N.D.
10	N.D.
20	0.12 ± 0.05 ^d^
30	1.81 ± 0.16 ^c^
40	3.04 ± 0.57 ^b^
45	3.43 ± 0.83 ^a^

Values are shown as the mean ± S.D. (*n* = 3). Different superscript letters indicate a significant difference (*p* < 0.05). (N.D.= Not detected).

**Table 4 foods-12-01584-t004:** Texture profile analysis of black garlic jam with different ratios by HPP treatment.

Groups	Hardness (g)	Fracturability (g)	Adhesiveness (g*s)	Springiness (s/s)
1:1.5:2	102.78 ± 0.12 ^e^	102.83 ± 0.21 ^f^	−215.01 ± 0.10 ^a^	0.96 ± 0.02 ^bc^
1:2:1	612.98 ± 0.18 ^b^	613.00 ± 0.22 ^b^	−1012.44 ± 0.64 ^d^	1.05 ± 0.03 ^a^
1:1:2	762.87 ± 74.59 ^a^	726.41 ± 49.47 ^a^	−367.29 ± 7.62 ^b^	0.92 ± 0.08 ^c^
1:1:1	491.90 ± 3.25 ^c^	491.90 ± 2.75 ^d^	−1112.00 ± 0.02 ^e^	1.04 ± 0.03 ^ab^
1:2:1.5	616.86 ± 0.42 ^b^	573.70 ± 0.07 ^c^	−1398.76 ± 0.15 ^f^	1.09 ± 0.02 ^a^
1:1.5:1	171.00 ± 0.12 ^d^	170.95 ± 0.12 ^e^	−450.47 ± 0.08 ^c^	1.06 ± 0.02 ^a^
Groups	Chewiness (g)	Gumminess (g)	Cohesiveness	Resilience
1:1.5:2	81.89 ± 0.21 ^e^	82.03 ± 0.15 ^e^	0.80 ± 0.02 ^bc^	0.03 ± 0.01 ^d^
1:2:1	635.63 ± 0.48 ^a^	611.88 ± 0.30 ^a^	1.01 ± 0.03 ^a^	0.07 ± 0.02 ^b^
1:1:2	135.59 ± 13.65 ^d^	322.38 ± 4.61 ^c^	0.62 ± 0.19 ^d^	0.05 ± 0.01 ^bc^
1:1:1	405.04 ± 0.15 ^c^	403.95 ± 0.17 ^b^	0.87 ± 0.01 ^b^	0.04 ± 0.01 ^cd^
1:2:1.5	440.98 ± 0.07 ^b^	403.83 ± 0.13 ^b^	0.67 ± 0.02 ^cd^	0.23 ± 0.01 ^a^
1:1.5:1	153.02 ± 0.10 ^d^	144.99 ± 0.02 ^d^	0.86 ± 0.01 ^b^	0.04 ± 0.01 ^cd^

Each value is expressed as mean ± S.D. (*n* = 40). Different superscript letters indicate a significant difference (*p* < 0.05). GE: glucose equivalent. The ratios of black garlic/water/sugar (*w/w*) included six groups: black garlic/water/sugar (*w/w*) = 1:1.5:2; black garlic/water/sugar (*w/w*) = 1:2:1; black garlic/water/sugar (*w/w*) = 1:1:2; black garlic/water/sugar (*w/w*) = 1:1:1; black garlic/water/sugar (*w/w*) = 1:2:1.5; and black garlic/water/sugar (*w/w*) = 1:1.5:1.

## Data Availability

The data presented in this study are available on request from the corresponding author.

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
