# Peer review of "Optimization of the Black Garlic Processing Method and Development of Black Garlic Jam Using High-Pressure Processing"

_foods, 2023, doi:10.3390/foods12081584_

Round 1

Reviewer 1 Report

Within this investigation, black garlic was studied, introducing many beneficial effects of the material, but showing its less spicy flavor. The aging condition and related products are still needed to be further investigated. The present study aims to analyze the beneficial effects under different processing condition and utilize high pressure processing in black garlic jam production.

Thematically the work is interesting for the researchers and professionals and the proposed manuscript is relevant to the scope of the journal. 

The manuscript should be rearranged and presented in the jouirnal's template. Please check the mentioned comments in the attachment. Several tzping errors should be corrected in the text.

The title is a clear representation of the manuscript's content.

The overall organization and structure of the manuscript are appropriate. The paper is well written and the topic is appropriate for the journal.

The aim of the paper is well described and the discussion was well approached, its results and discussion are correlated to the cited literature data.

The literature review is comprehensive and properly done. Perhaps a few newer references could be introduced (many references are from previous century), especialy in Introduction and Discussion sections?

The novelty of the work must be more clearly demonstrated. The results of ANOVA analysis is somehow not presented, which variable influenced more to the total antioxidants activity or reducing power of hot water extract?

Statistical interpretation of the analytical data must be more properly presented.

Other Specific Comments: The work is properly presented in terms of the language. The work presented here is very interesting and well done, it is presented in a compact manner.
In general, there are no doubtful or controversial arguments in the manuscript. The methodology applied in the research is presented in clear manner, so that it is repeatable by other authors.
The results are presented in a logical sequence and the discussion and analysis of the results are properly elaborated. 

Perhaps more details could be introduced in section 3. Results and discussion and especially in section 4. Conclusion. Perhaps more numerical data, the results of this study could be presented in section 4.

The main drawback of the paper i s the extent of novelty, or the main novelty in the present work, compared to the works of other researchers? In my opinion, the authors should put additional effort to demonstrate that the present work gives a substantial contribution in the research area.

Reviewer 2 Report

1.- All the figures and most of the tables are of low quality, they should be improved

2.- From reference 31, the years of the references must be in bold

3.- Address the observations that are found as comments in the attached file

4.- Review and correct the homogeneity of the format of the references.

5.- All the chromatograms of figure 4 should be put on a single graph, so that the differences and the increase of 5-HMF can be better observed

6.-There is no homogeneity in the abbreviations, you have to review them all.

Reviewer 3 Report

Lines 83-84. The fresh garlic was processed in the same way... Remark: This sentence should be clarified, as follows, “The fresh garlic was extracted in the same way”....

Section 2.2. Remark: In what form was the garlic processed and extracted, was it in the form of cloves or was the garlic pre-cut?

It seems more logical if section 2.5. “Preparation of Black Garlic Jam”... will be moved up, after section 2.2. “Black garlic Sample Preparation”.

Remark regarding the high pressure of 300 MPa used for the preparation of black garlic jam: The authors must justify why just the pressure of 300 MPa was chosen for the preparation of black garlic jam, and whether this pressure is optimal? The same regarding the time of pressure 15 min - the authors must justify this time.

Lines 172-173. Figure 1C and Table 172 1 present that the longer aging duration increased moisture... Remark: It is an error because from Figure 1C it follows that with an increase in aging duration, moisture content decreases.

Lines 174-175. The yield of black garlic aged for 5, 10, 20, 30, 40, and 45 days was 63.85%, 67.36%, 68.93%, 58.43%, 52.71%, and 49.23%,... Remark: Too many unnecessary enumerations. I recommend correcting this sentence as follows, “The yield of extract from black garlic increased from 63.8 to 68.9% with an increase in aging duration from 5 to 20 days, but further, a decrease in the yield was observed. For comparison, the yield of extract from fresh garlic was 33.7% (Table 1).

Remark regarding Table 1: Since the decrease in water activity (Aw) is a consequence of the extraction, this Aw-column should be moved and placed after the extract yield column.

Lines 181-183. Remark: Too many unnecessary enumerations. I recommend correcting this sentence as follows, “Figure 2A illustrates that DPPH scavenging effect (5 mg/ml) of fresh garlic was 11.2% only. For aged black garlic, this effect increased from 40.4 to 86.2% with the increase in aging duration from 5 to 30 days but further slightly decreases”.

Lines 210-212. Remark: Too many unnecessary enumerations. I recommend correcting this sentence as follows, “The results depict that TEAC (5 mg/ml) of fresh garlic was 25.3%. TEAC of aged black garlic increased from 33.2 to 88.4% with the increase in aging duration from 5 to 30 days and further decreases”.

Lines 229-230. I recommend correcting this sentence as follows, “Total phenols (at mg GAE/g dw) in the extract of black aged garlic increased from 10.9 to 74.9% with the increase in aging duration from 5 to 30 days and further decreases”.

 Lines 248-249. I recommend correcting this sentence as follows, “... while the reducing sugar in the extract of black aged garlic increased from 10 to 39 when the aging duration increased from 5 to 30 days, but further decreases”.
